# Chloroplast Genome of Rambutan and Comparative Analyses in Sapindaceae

**DOI:** 10.3390/plants10020283

**Published:** 2021-02-02

**Authors:** Fei Dong, Zhicong Lin, Jing Lin, Ray Ming, Wenping Zhang

**Affiliations:** 1College of Life Sciences, Fujian Agriculture and Forestry University, Fuzhou 350002, China; dongfei9508@163.com or; 2College of Agriculture, Fujian Agriculture and Forestry University, Fuzhou 350002, China; woshi0655@126.com or; 3Center for Genomics and Biotechnology, Fujian Provincial Key Laboratory of Haixia Applied Plant Systems Biology, Key Laboratory of Genetics, Fujian Agriculture and Forestry University, Fuzhou 350002, China; lolyemily@163.com or; 4Department of Plant Biology, University of Illinois at Urbana-Champaign, Urbana, IL 61801, USA

**Keywords:** rambutan, *Nephelium lappaceum*, chloroplast genome, Sapindaceae, comparative genomic, RNA editing, phylogeny

## Abstract

Rambutan (*Nephelium lappaceum* L.) is an important fruit tree that belongs to the family Sapindaceae and is widely cultivated in Southeast Asia. We sequenced its chloroplast genome for the first time and assembled 161,321 bp circular DNA. It is characterized by a typical quadripartite structure composed of a large (86,068 bp) and small (18,153 bp) single-copy region interspersed by two identical inverted repeats (IRs) (28,550 bp). We identified 132 genes including 78 protein-coding genes, 29 tRNA and 4 rRNA genes, with 21 genes duplicated in the IRs. Sixty-three simple sequence repeats (SSRs) and 98 repetitive sequences were detected. Twenty-nine codons showed biased usage and 49 potential RNA editing sites were predicted across 18 protein-coding genes in the rambutan chloroplast genome. In addition, coding gene sequence divergence analysis suggested that *ccsA*, *clpP*, *rpoA*, *rps12*, *psbJ* and *rps19* were under positive selection, which might reflect specific adaptations of *N. lappaceum* to its particular living environment. Comparative chloroplast genome analyses from nine species in Sapindaceae revealed that a higher similarity was conserved in the IR regions than in the large single-copy (LSC) and small single-copy (SSC) regions. The phylogenetic analysis showed that *N. lappaceum* chloroplast genome has the closest relationship with that of *Pometia tomentosa*. The understanding of the chloroplast genomics of rambutan and comparative analysis of Sapindaceae species would provide insight into future research on the breeding of rambutan and Sapindaceae evolutionary studies.

## 1. Introduction

Rambutan (*Nephelium lappaceum* L.) is an important tropical fruit in the family Sapindaceae and originated in Indonesia and the Malay Peninsula [1]. It is widely cultivated in Southeast Asia and the coastal areas of South China. Malaysians refer to it as “rambutan”, because the fruit surface is covered with thick and elongated spines. The fruits of rambutan are popular in the general population due to their rich nutrients, delicate and characteristic flavor and delicious taste. Rambutan peel extract is rich in phenolic content and exhibited antibacterial activity against many pathogenic bacteria, suggesting its antioxidant and/or antimicrobial properties [2]. Rambutan has the potential to be used in natural antioxidants and anti-aging agents in pharmaceutical and food industries to replace synthetic ones [3,4].

The Sapindaceae family contains over 150 genera and 2000 species with several economically important crops, which are widely distributed in tropical and subtropical regions [5]. However, genomic research on the Sapindaceae family, especially in the *N. lappaceum*, has been relatively scarce. This lack of genetic information makes it difficult to meet the need for improving the quality and agronomic characteristics of rambutan through breeding and gene editing.

Chloroplast (cp) are photosynthetic organelles that provide energy to green plants; they play an important role in the photosynthesis and secondary metabolic activities [6,7]. The chloroplast genomes are maternally inherited in most plants and are highly conserved in terms of their composition and sequence. The typical chloroplast genomes of angiosperms are circular DNA molecules which have a characteristic quadripartite structure with a large single-copy (LSC) region, a small single-copy (SSC) region, and two inverse repeats (IRs) regions [6]. The length of the genome is between 120 and 170 kb and usually encodes 110 to 130 genes, and about 40 genes are specialized, participating in photosynthesis, transcription and translation [8,9]. The first chloroplast genome from tobacco (*Nicotiana tabacum* L.) was sequenced in 1986 [10], With the rapid development of next-generation sequencing technologies, the cost of whole genome sequencing is dropping rapidly [11]. Complete chloroplast genome sequences now could be easily acquired with a relatively low cost. There has been an explosion in the number of available chloroplast genome sequences. Over 5000 complete chloroplast genome sequences have been submitted in the National Center for Biotechnology Information (NCBI) organelle genome database. 

Within the Sapindaceae family, the complete chloroplast genomes of eight plant species have been sequenced and were available from the NCBI database. Nevertheless, no chloroplast genome in the genus *Nephelium* L. has been reported. In this study, we report the first complete chloroplast genome of *N. lappaceum*, exploring its general features, SSRs and long repeats, codon usage and analysis of IR contraction and expansion. In addition, nine chloroplast genome sequences were used for the analysis of molecular evolution in the Sapindaceae family. We constructed a phylogenetic tree to understand the phylogenetic relationships of Sapindaceae plants. The chloroplast genome sequence and the comprehensive chloroplast genomic analysis of *N. lappaceum* would provide a theoretical basis for molecular identification and further understanding of the evolutionary history of the Sapindaceae family.

## 2. Results

### 2.1. Chloroplast Genome Features of N. lappaceum

The structure of the *N. lappaceum* chloroplast genome was analogous to most chloroplast genomes of plants with a typical quadripartite structure. We assembled a closed circular chloroplast genome with 161,321 bp in *N. lappaceum*. The chloroplast genome contains a pair of inverted repeat regions (IRs) of 28,550 bp, a large single-copy region (LSC) of 86,068 bp and a small single-copy region (SSC) of 18,153 bp (Figure 1). Besides, the overall nucleotide composition of rambutan is: 30.79% A, 31.44% T, 19.27% C, and 18.50% G, with a total GC content of 37.77%. In total, 132 genes were annotated on this chloroplast genome, including 78 protein-coding genes, 29 transfer RNA genes (tRNA) and 4 ribosomal RNA genes (rRNA). Among them, a total of 21 genes were found duplicated in the IR regions, including nine protein-coding genes (*rps3*, *rps7*, *rps12*, *rps19*, *rpl2*, *rpl22*, *rpl23*, *ndhB* and *ycf2*), eight tRNA genes (*trnA-UGC*, *trnI-CAU*, *trnI-GAU*, *trnL-CAA*, *trnM-CAU*, *trnN-GUU*, *trnR-ACG* and *trnV-GAC*) and four rRNA genes (*rrn4.5s*, *rrn5s*, *rrn16s* and *rrn23s*) (Appendix A). The gene structure analysis showed that 21 genes contain introns, and 19 of them (11 protein-coding genes and 8 tRNA genes) have one intron, while two genes (*ycf3* and *clpP*) have two introns (Appendix A). We characterized the basic features of the chloroplast genome in Sapindaceae and compared them with *N. lappaceum*. The size of the *N. lappaceum* chloroplast genome was slightly larger than that of *Sapindus mukorossi* Gaertn. (160,481 bp), *Pometia tomentosa* (Blume) Teijsm. & Binn. (160,818bp), *Dimocarpus longan* Lour. (160,833bp), *Dodonaea viscosa* (Linn.) Jacq. (159,375bp), and *Eurycorymbus cavaleriei* (H.Lev.) Rehder & Hand.-Mazz. (158,777bp), but shorter than that of *Litchi chinensis* Sonn. (162,524 bp), *Koelreuteria paniculata* Laxm. (163,258bp), and *Xanthoceras sorbifolium* Bunge. (161,231bp) chloroplast genome of Sapindaceae (Table 1). The number of chloroplast genes in *N. lappaceum* was 132, the same as that of *D. longan*, *L. chinensis* and *X. sorbifolium* (Table 1). The highest number of genes was found in E. *cavaleriei* (137), followed by *S. mukorossi* (135) and *D. viscosa* (135). The number of protein-coding genes and tRNA genes in these species varied from 85 to 89 and 37 to 40, respectively. Furthermore, there was no significant difference in GC content among the nine analytical genomes in Sapindaceae. 

### 2.2. Characterization of SSRs and Repeat Sequences

A total of 63 SSRs were detected from rambutan chloroplast genome, of which 45 were mononucleotide, 3 dinucleotide, 8 trinucleotide, 5 tetranucleotide and two pentanucleotide (Figure 2B). Moreover, we compared the distribution pattern and number of SSRs with eight other chloroplast genomes in the Sapindaceae family (Appendix A, Appendix A). The number of mononucleotide repeats was more than the sum of other types (Figure 2A), and the number and types of chloroplast SSRs varied in different species. *S. mukorossi* (91 SSRs) possesses the highest number of SSRs while *E. cavaleriei* (62 SSRs) possesses the lowest. Furthermore, the chloroplast genome of *D. longan*, *L. chinensis*, *P. tomentosa*, *D. viscosa*, *K. paniculata* and *X. sorbifolium* contained 79, 75, 74, 77, 87 and 83 SSRs, respectively. In this study, a total of 98 larger repeats (>10 bp) were identified in *N. lappaceum* chloroplast genome, composed of 42 forward, 11 reverse, 41 palindromic and 4 complement repeats (Appendix A) using REPuter [12]. Among them, the largest repeat was a palindromic repeat with a size of 48 bp.

### 2.3. Codon Usage Analysis and RNA Editing Sites Prediction

We used 53 protein-coding sequences from rambutan chloroplast genome to calculate codon usage frequency and relative synonymous codon usage (RSCU) frequency (Appendix A). All protein-coding sequences contain 21,434 codons. In detail, leucine and cysteine were the highest and lowest number of amino acids; they had 2232 codons (approximately 10.41% of the total) and 236 codons (approximately 1.10% of the total), respectively. While Met (ATG) and Trp (TGG) were encoded by only one codon, showing no biased usage (RSCU = 1). There are 30 codons with RSCU values greater than 1, indicating that they showed biased usage (Figure 3). Among them, excluding the leucine (UUG) codon, which was G-ending, the remaining 29 biased usage codons of *N. lappaceum* were all A/T-ending in the third codon. The usage was generally biased towards A or T(U) with higher RSCU values, including UUA (1.91) in leucine, the stop-codon UAA (1.87), and GCU (1.74) in Alanine in the chloroplast genome of *N. lappaceum*. Besides, there were 49 potential RNA editing sites found across 18 protein-coding genes in the *N. lappaceum* chloroplast genome, and the *ndhB* gene contained the most RNA editing sites (9) (Appendix A). We also observed that RNA editing sites all showed C to U conversion, which took place at the first (30.6%) or second (69.4%) positions of the codons, indicating that editing in the third codon position occurred at lower frequency than that in the second or first codon position. Furthermore, serine codons were more frequently edited than codons of other amino acids and the conversion from serine to leucine occurred most frequently.

### 2.4. Comparative Genomes Analysis

The comparative analysis based on mVISTA was performed between the chloroplast genomes of rambutan, with the other eight Sapindaceae species with the annotated *D. longan* chloroplast genome as a reference. The nine Sapindaceae family chloroplast genomes’ length was between 158,777 and 163,258 bp. The chloroplast genome of *K. paniculata* had the largest size, whereas *E. cavaleriei* had the smallest size. Interestingly, the SSC region (16,568 bp) of *L. chinensis* was the shortest, whereas the SSC region (18,873 bp) of the *S. mukorossi* chloroplast genome was the longest (Figure 4). The IR (A/B) regions exhibited less divergence than the SSC and LSC regions. In addition, the coding regions were more highly conserved than the non-coding regions. Among the nine chloroplast genomes, four rRNA genes (*rrn16S*, *rrn23S*, *rrn5S*, *rrn4.5S*) were the most conserved, while four genes (*ndhF*, *ndhD*, *ndhH* and *ycf1*) showed the most diversity in the coding regions. The highly divergent regions were found in the intergenic spacers and introns, including *trnH-GUG-psbA*, *trnS-GCU-trnG-UCC*, *trnR-UCU-atpA*, *atpF-atpH*, *petN-psbM*, *psbZ-trnG-GCC*, *trnF-GAA-ndhJ*, *ndhC-trnV-UAC*, *psbE-petL*, *ndhF-rpl32*, *rpl16-rps3* and *rpl32-trnL-UAG*.

### 2.5. Expansion and Contraction of IR Regions

We compared the IR regions and the junction sites of the LSC and SSC regions of nine Sapindaceae family chloroplast genomes (including *N. lappaceum*) (Figure 5). The IR regions vary in different chloroplast genomes, ranging from 26,923 bp in *E. cavaleriei* to 30,103 bp in *L. chinensis*. In our study, the *ycf1* gene was located at the SSC/IRA junction in all of the nine chloroplast genomes and the fragment located at the IRa region ranged from 962 bp to 3183 bp. Moreover, most junctions between LSC and IRa in this study were located downstream of the *trnH-GUG*, except the *S. mukorossi*. In addition, the LSC/IRb junction of three species *D. viscosa*, *E. cavaleriei* and *K. paniculata* was located within the coding region of *rpl22* and created a location of 110, 40 or 63 bp at the LSC/IRb border. The remaining chloroplast genomes share a similar pattern; the LSC/IRb junction was located in intergenic regions of *rpl16* and *rps3*, and the IRb/SSC junction between IRb and SSC region (JSB) of five species (*S. mukorossi*, *X. sorbifolium*, *D. viscosa*, *E. cavaleriei* and *K. paniculata*) was located between the gene of *ycf1* and *ndhF*. However, the other four chloroplast genomes only have *ndhF* located at or near the JSB.

### 2.6. Synonymous (Ks) and Non-Synonymous (Ka) Substitution Rate Analysis

To explore the molecular evolution of orthologous genes shared by nine Sapindaceae species, particularly genes undergoing purifying or positive selection, we calculated the Ka/Ks ratio of 622 orthologous pairs with 78 protein-coding genes (Appendix A). Overall, the average Ka/Ks ratio of the nine chloroplast genomes was 0.20. In total, 612 orthologous pairs had a Ka/Ks ratio of less than 1 in the nine comparison groups, out of which 546 orthologs had a Ka/Ks ratio of less than 0.5 (Figure 6), suggesting that most genes were undergoing strong purifying selection pressures. Moreover, 66 orthologs of 31 genes with a Ka/Ks ratio between 0.5 and 1, and 10 orthologous pairs of 6 genes (*ccsA*, *rpoA*, *rps12*, *psbJ*, *clpPc* and *rps19*) with a Ka/Ks ratio greater than 1, were detected in this study, suggesting that these genes might have experienced positive selection in the procedure of evolution. Among them, the Ka/Ks ratio of the *ycf1* gene was greater than 0.5 in eight comparison groups; the *rpoA* and *ycf2* gene with a Ka/Ks ratio greater than 0.5 was also observed in the comparison of seven and six groups, respectively. Besides, the *clpP*, *matK* and *rps15* genes, with Ka/Ks ratios >0.5, were found in four out of the eight comparison groups.

### 2.7. Phylogenetic Analysis

We performed multiple sequence alignments using the whole chloroplast genome sequences of nine Sapindaceae species and two Anacardiaceae species as outgroups (Figure 7). All nodes in the ML trees have 100% bootstrap support values, and these 11 chloroplast genome sequences were clustered into three groups. In detail, the five species (*D. longan*, *L. chinensis*, *P. tomentosa*, *N. lappaceum* and *S. mukorossi*) from Sapindoideae clustered into one group, four species (*K. paniculata*, *D. viscosa*, *E. cavaleriei* and *X. sorbifolium*) from Dodonaeoideae were in one group, and the two species (*A. occidentale* and *M. indica*) in Anacardiaceae were clustered into one group. In the Sapindoideae group, the *N. lappaceum* chloroplast genome sequence showed the closest relationship with *P. tomentosa*, followed by *D. longan* and *L. chinesis*, as far as *S. mukorossi*. The three groups of this phylogenetic tree of the 11 chloroplast genome sequences were consistent with traditional taxonomy, suggesting that the chloroplast genome could effectively resolve the phylogenetic positions and relationships of species.

## 3. Discussion

We assembled the complete *N. lappaceum* chloroplast genome sequence and deposited it to GenBank under accession number: MT936934. The *N. lappaceum* chloroplast genome is consistent with the characteristics of most angiosperm species in structure and gene content. Although there are some differences in the sizes of the overall genome, LSC, SSC and IR regions, the numbers of genes and GC content are similar among the nine Sapindaceae chloroplast genomes, which, to some extent, reflects the high conservation of angiosperm chloroplast genomes [6]. Notably, the number of tRNA genes in *E. cavaleriei* and *S. mukorossi* are quite different from other species in Sapindaceae since some of the tRNA genes types and copy numbers are different. The copy number of tRNA genes may be affected by differences in gene codon composition and amino acid usage [13]. Intron plays an important role in RNA stability, regulation of gene expression and alternative splicing, which has been reported in many other species [14,15]. There were two genes (*ycf3* and *clpP*), in the *N. lappaceum* chloroplast genome, that included two introns. It has been reported that the *ycf3* gene was essential for the accumulation of the photosystem I (PSI) complex and acts as a chaperone that interacts with the PSI subunits at a post-translational level [16,17]. Besides, the *clpP* gene functions as the proteolytic subunit of the ATP-dependent Clp protease in plant chloroplasts and has been shown to be essential for the development and/or function of plastids with active gene expression in previous studies [17,18,19]. Thus, the study of the *ycf3* and *clpP* genes will contribute to further investigation of chloroplast in *N. lappaceum*.

Simple sequence repeats (SSRs), also known as microsatellites, are tandem repeats distributed across the entire genome which have been widely applied as molecular markers for determining genetic variations across species in evolutionary studies because of their unique uniparental inheritance [20,21,22]. The mononucleotide SSRs, A and T, were identified most frequently (68% on average) among the nine analyzed chloroplast genomes of Sapindaceae species. This result is consistent with the previous report that poly A and T are the most abundant repeats in most angiosperm chloroplast genomes [23,24,25]. Moreover, repetitive sequences are helpful in phylogenetic study and play a vital role in genome rearrangement [25]. Most of the repetitive sequences in the *N. lappaceum* chloroplast genome are distributed in IGS regions, whereas few are located in the region of the protein-coding genes. These results can provide chloroplast molecular markers of family Sapindaceae that can be used to quickly identify species and confirm hybrid progeny when breeding.

Codon usage biases are found in all eukaryotic and prokaryotic genomes and have been proposed to regulate different aspects of the translation process [26]. High RSCU values of the codons are probably attributed to amino acid functions or peptide structures that avoid transcriptional errors in chloroplast genomes [27,28]. The phenomenon of codons ending in A/T in our study is similar to the pattern reported for other chloroplast genomes, which may be caused by a composition bias for a high A/T ratio [23]. High codon preference is prevalent in other land plant chloroplast genomes and the results of our study are similar to those of other species with chloroplast genome codon usage biases. The research on codon preferences can help us to better understand the gene expression and molecular evolution mechanisms of *N. lappaceum*.

We observed that the *ndhB* gene contained the most RNA editing sites within the 49 potential RNA editing sites, and 16 editing sites were U_A type, indicating there was a U_A bias for the distribution of RNA editing sites that were in accordance with previous reports in other species [24,29]. RNA editing is a post-transcriptional regulation pattern involved in the insertion, deletion, or modification of nucleotides that widely exists in land plants [30]. The first chloroplast RNA editing event of a land plant was discovered in the mRNA transcript of the *rpl2* gene in the maize chloroplast genome in 1991 [31]. The most frequent editing events in plants are C-to-U changes; however, U-to-C editing has also been observed [32,33]. Additionally, RNA editing usually occurs in the first or second base of codons, resulting in the conversion of hydrophilic amino acid to hydrophobic [34].

Comparative analysis of chloroplast genomes is an essential step in genomics that can provide insight into complex evolutionary relationships. The mVISTA analysis showed that nine Sapindaceae chloroplast genomes were conserved, with a high degree of similarity and gene order conservation, and the coding region was more conserved than the non-coding region, which is consistent with reports on other angiosperms [35], suggesting an evolutionary conservation of these genomes at the genome-scale level. Notably, the five species in the Sapindoideae subfamily presented the same divergence pattern, but the divergence of the four species in the Dodonaeoideae subfamily was greater. In addition, the *ycf1* gene showed the greatest degree of differentiation. Previous studies reported that *ycf1* is helpful to provide phylogenetic information at the species level and more variable than *matK* in Orchidaceae [36]. Furthermore, *ycf1* performed better in identifying DNA barcodes of high resolution at the species level than either *matK*, *rbcL* or *trnH-psbA* [35]. These variable genic regions found in our study can be regarded as molecular markers for DNA barcoding and phylogenetic studies in Sapindaceae.

Although most land plants have relatively conserved cp genomes, the end of the inverted repeats (IRa and IRb) regions differs among various plant species. The expansion or contraction of the IR regions represent important evolutionary events that often influence the size variation of different chloroplast genomes, and it is thus helpful to study the chloroplast genome evolution history [37,38]. In this study, our results suggested that the boundary of IR/LSC and IR/SSC might be conserved among chloroplast genomes of closely related family species but some differences also occur between relatively distantly related family species, such as gene overlap length, a duplicate of the *ycf1* and *rps3* genes, even the distance of *trnH-GUG* from the border near the LSC/IRB junctions, indicating that the expansion and contraction of the IR regions led to length and structure changes in chloroplast genomes.

The ratio between nonsynonymous (Ka) and synonymous (Ks) nucleotide substitution has been widely used as an important marker in genome or gene evolution studies [8]. Ka/Ks = 1 signifies neutral evolution, Ka/Ks > 1 indicates that the gene is affected by positive selection, whereas Ka/Ks  <  1 indicates that the gene is affected by purifying selection [39]. Additionally, a Ka/Ks ratio of 0.5 was considered as a useful cut-off value to identify genes under positive selection in previous studies [40]. In our study, the *ccsA*, *rpoA*, *rps12*, *psbJ*, *clpP* and *rps19* gene with Ka/Ks > 1. It is noteworthy that the *ycf1* gene also exhibited high Ka/Ks ratios with Ka/Ks > 0.5 in eight comparison groups. This result is in keeping with the previous observations that the *ycf1* gene was more variable than the *matK* and *rbcL* genes in most plants, and could be used as an effective biological tool for plant phylogeny study [35]. The positive selection of genes in *N. lappaceum* possibly provided help for adaptations to its particular living environment.

Numerous studies have shown that chloroplast genome sequences have been successfully used in taxonomic and phylogenetic studies [41], and contribute to describing the evolutionary relationships between species [42]. In this study, the topology of the trees consists of two main branches: Dodonaeoideae and evolutionary younger Sapindoideae. Furthermore, generic relationships of the two subfamilies are basically congruent with the taxonomy of these families. The availability of the completed *N. lappaceum* chloroplast genome provided us with sequence information that can be used to confirm the phylogenetic position of *N. lappaceum* and understand the phylogenetic relationships among Sapindaceae. However, as we used only a small number of species in Sapindaceae, further research on other chloroplast genomes as well as nuclear genome sequences of Sapindaceae should be conducted to provide more sufficient evidence to accurately illustrate the evolution of the family Sapindaceae.

## 4. Materials and Methods

### 4.1. Plant Material, DNA Extraction, and Sequencing

Young, healthy leaves of the major cultivar of rambutan, Baoyan7, were collected from Baoting (N18°23′, E109°21′) in Hainan Province, China. The leaves were frozen in liquid nitrogen and maintained at −80 °C. The total genomic DNA was extracted by 2X cetyltrimethylammonium bromide (CTAB) method [43]. In addition, a library with insert sizes of 300–500 bp was constructed and then sequenced on an Illumina HiSeq2500 platform (Illumina, San Diego, CA, USA) using the double terminal sequencing method (150 pair-ends).

### 4.2. Chloroplast Genome Assembly and Annotation

First, FastQC software was performed to evaluate the quality of Illumina paired-end raw reads [44], and low-quality reads were filtered. The remaining clean reads were used for assembly via NOVOPlasty [45] using *D. longan* chloroplast genome(GenBank: MG214255) [46] as the reference genome to generate the first version of rambutan genome. Next, all clean reads were mapped onto the first version genome and the mapped reads were assembled using SPAdes3.14.1 [47] and assembled contigs were corrected using the pair-end short reads from HiSeq2500 in Pilon version 1.23 (https://github.com/broadinstitute/pilon) [48] to generate the second version of rambutan chloroplast genome. These two versions were compared and mutually corrected to get the final complete rambutan chloroplast genome.

The chloroplast genome was annotated by the online program GeSeq (https://chlorobox.mpimp-golm.mpg.de/geseq.html) [49] and CPGAVAS2 [50]. Genome features like start/stop codons and intron/exon borders were manually corrected through the comparison of other reported Sapindaceae family chloroplast genomes. In addition, tRNA genes were identified by tRNAscan-SE 2.0 (http://lowelab.ucsc.edu/tRNAscan-SE/) [51]. A circular map of the revised annotated rambutan chloroplast genome was illustrated by using Organellar Genome DRAW (OGDRAW) (https://chlorobox.mpimpgolm.mpg.de/OGDraw.html) [52].

### 4.3. Chloroplast Genome Analysis

The simple sequence repeats (SSR) in nine chloroplast genome sequences of Sapindaceae (including *N. lappaceum*) (Appendix A) were identified using the MISA online tool (https://webblast.ipk-gatersleben.de/misa/) [53] and the threshold settings were as follows: ten was applied to mononucleotide repeats, five to dinucleotide repeats and four to trinucleotide repeats, three for tetra-, penta-, and hexanucleotide repeats [54]. Repetitive sequences including forward, reverse, palindrome, and complement sequences were analyzed by REPuter program [12], and the parameter was set with the minimum length of repeat region set to 10bp and the minimum sequence identity set to 90%. 

The expansion and contraction of the inverted repeat (IR) regions at junction sites from eight Sapindaceae family chloroplast genome sequences, including *Dimocarpus longan* (MG214255), *Litchi chinensis* (KY635881), *Pometia tomentosa* (MN106254), *Sapindus mukorossi* (KM454982), *Dodonaea viscosa* (MF155892), *Eurycorymbus cavaleriei* (MG813997), *Koelreuteria paniculata* (KY859413) and *Xanthoceras sorbifolium* (KY779850), were examined and plotted using IRscope online program (https://irscope.shinyapps.io/irapp/) [55]. Codon usage of the *N. lappaceum* chloroplast genome was analyzed via the GALAXY platform (https://galaxy.pasteur.fr) [56] with the CodonW online tool. Protein-coding genes of less than 300 nucleotides in length, and the repetitive gene sequences, were removed to reduce deviation of the results [57]. Finally, 53 CDS in *N. lappaceum* were selected for further codon usage analysis. Besides, putative RNA editing sites were predicted using the PREP-Cp web server (http://prep.unl.edu/cgi-bin/cp-input.pl) [58] with a cutoff value of 0.8.

### 4.4. Genome Comparison

We downloaded four whole chloroplast genome sequences of Sapindaceae family from the National Center for Biotechnology Information (NCBI) Organelle Genome and Nucleotide Resources database, including *D. longan* [46], *L. chinensis*, *P. tomentosa* [59], *S. mukorossi* [60], *D. viscosa* [61,62], *E. cavaleriei* [62], *K. paniculata* [63] and *X. sorbifolium* [64]. The mVISTA online program (Shuffle-LAGAN mode) [65,66] was used to compare chloroplast genome sequence of rambutan with the other species from Sapindaceae, in which the annotation of *D. longan* was the reference.

### 4.5. Positive Selection Analysis of Protein Sequence

We analyzed synonymous (Ks) and non-synonymous (Ka) substitution rates to investigate the molecular evolutionary process of the Sapindaceae family, The protein-coding genes of *N. lappaceum* were separately compared with eight closely related species in the Sapindaceae family: *D. longan*, *L. chinensis*, *P. tomentosa*, *S. mukorossi*, *D. viscosa*, *E. cavaleriei*, *K. paniculata* and *X. sorbifolium*, using ParaAT 2.0 [67], then the Ka/Ks value was calculated using KaKs_calculator 2.0 [68] with the NG method [69].

### 4.6. Phylogenetic Analysis

In order to deeply detect the evolutionary relationships of Sapindaceae family, we aligned 9 complete chloroplast genomes (including *N. lappaceum*) with MAFFT version 7 [70]. The best fitting nucleotide substitution model (TVM + I + G) was chosen by jModelTest v2.1.7 [71]. Phylogenetic analysis was then inferred by ML (maximum-likelihood) method based on the TVM + I + G substitution model in PAUP* 4.0 [72] with 1000 bootstrap replicates. *Anacardium occidentale* (KY635877) and *Mangifera indica* (KY635882) [73] in Anacardiaceae family were set as the outgroup.

## 5. Conclusions

We assembled the first complete chloroplast genome of rambutan using Illumina sequencing technology and compared its structure with other Sapindaceae species. The chloroplast genome of *N. lappaceum* exhibits similar quadripartite structure, gene order, and G + C content, when compared with other Sapindaceae chloroplast genomes. A total of 63 SSRs and 98 repeat sequences were identified in the *N. lappaceum* chloroplast genome. The research on codon usage of *N. lappaceum* shows that some amino acids have obvious codon usage bias and the codon preferences may help us to understand the evolution mechanisms of *N. lappaceum*. With PREP prediction, we detected 49 RNA editing loci in 18 protein-coding genes in *N. lappaceum*. Moreover, the expansion and contraction of the IR regions led to variations in the genome sizes of nine Sapindaceae chloroplasts. There are 6 genes (*ccsA*, *rpoA*, *rps12*, *psbJ*, *clpP* and *rps19*) that were detected with a Ka/Ks ratio >1, suggesting that these genes experienced positive selection in the evolution. Additionally, phylogenetic analysis using nine complete chloroplast genome sequences in Sapindaceae strongly supports the close relationships of *N. lappaceum* and *P. tomentosa* among sequenced chloroplast genomes in Sapindaceae.

## Figures and Tables

**Figure 1 plants-10-00283-f001:**
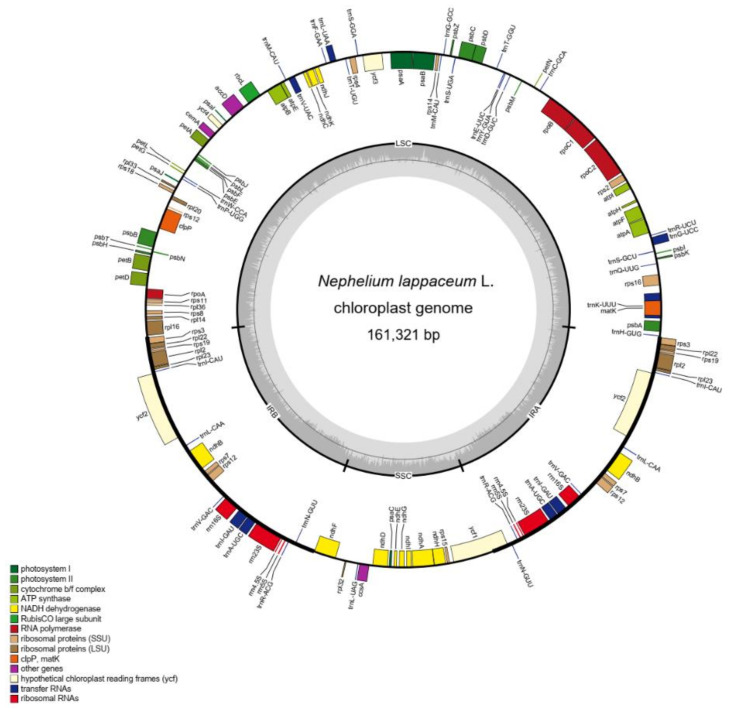
Gene map of *N. lappaceum* chloroplast genome. Genes drawn outside and inside of the circle are transcribed clockwise and counterclockwise, respectively. Genes belonging to different functional groups are color coded. The darker gray in the inner circle corresponds to GC content. Small single-copy (SSC) region, large single-copy (LSC) region, and inverted repeats (IRA and IRB) are indicated.

**Figure 2 plants-10-00283-f002:**
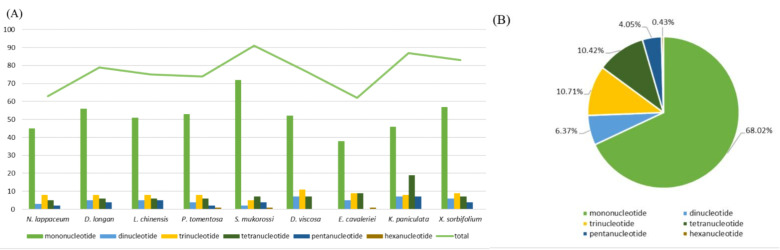
Analysis of simple sequence repeats (SSRs) in nine Sapindaceae (including *N. lappaceum*) chloroplast genomes. (**A**) Number of different SSRs types detected in nine Sapindaceae (including *N. lappaceum*) chloroplast genomes. (**B**) Presence of different SSRs types in all SSRs of nine Sapindaceae (including *N. lappaceum*) chloroplast genomes.

**Figure 3 plants-10-00283-f003:**
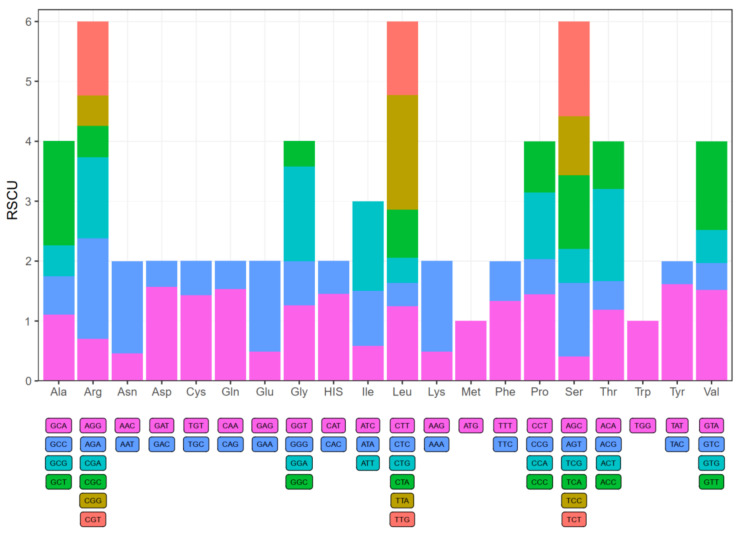
Codon content of 20 amino acids and stop codons in all protein-coding genes of *N. lappaceum* chloroplast genome. The colour of the histogram corresponds to the colour of codons.

**Figure 4 plants-10-00283-f004:**
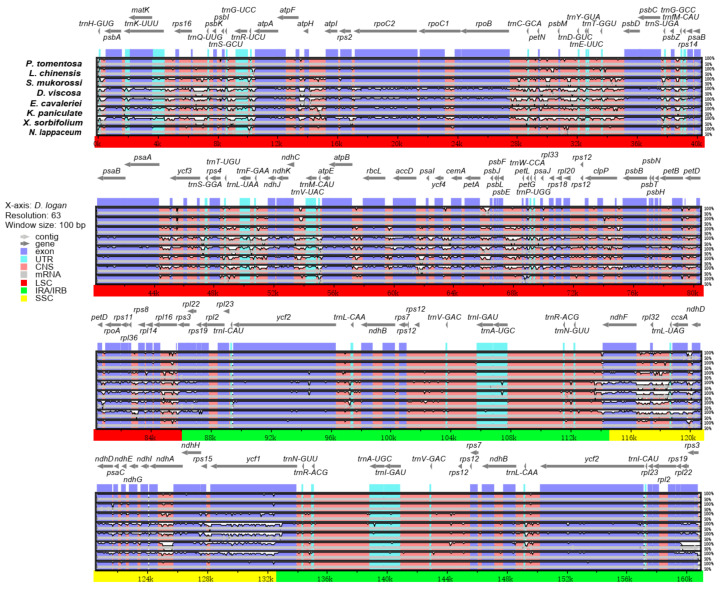
Comparison of nine Sapindaceae chloroplast genomes (including *D. longan*, *L. chinensis*, *P. tomentosa*, *S. mukorossi*, *D. viscosa*, *E. cavaleriei*, *K. paniculata*, *X. sorbifolium* and *N. lappaceum*), with *D. longan* as a reference. Gray arrows and thick black lines above the alignment indicate the direction of the gene. Purple bars represent exons, blue bars represent untranslated regions (UTRs), pink bars represent conserved non-coding sequences (CNS), and gray bars represent mRNA. The y-axis indicates the identity, expressed as a percentage, between 50% and 100%.

**Figure 5 plants-10-00283-f005:**
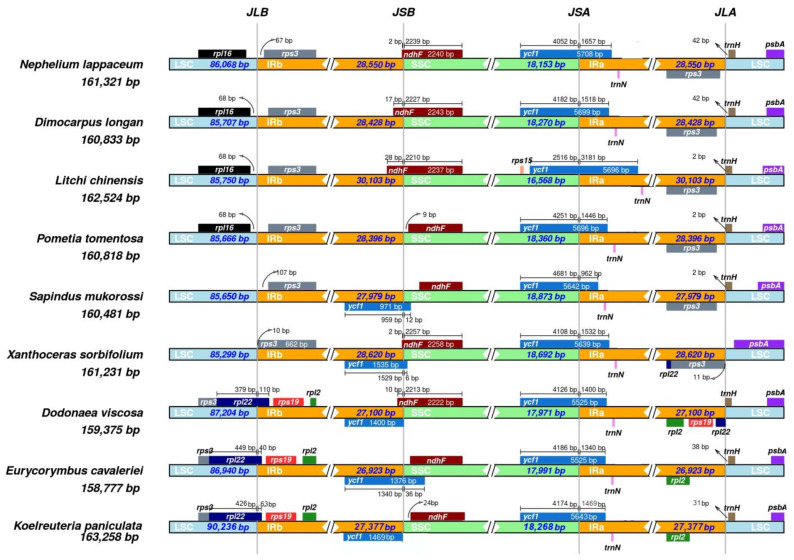
Comparison of the borders of the LSC, SSC and IR regions among nine Sapindaceae chloroplast genomes. For each species, genes transcribed in positive strand are depicted on the top of their corresponding track from right to left direction, while the genes on the negative strand are depicted below from left to right. The numbers at arrows refer to the distance of the start or end position of a given gene from the corresponding junction site. The T bars above or below the genes indicate the extent of their parts with their corresponding values in the base pair. The plotted genes and distances in the vicinity of the junction sites are the scaled projection of the genome. JLB (IRb /LSC), JSB (IRb/SSC), JSA (SSC/IRa) and JLA (IRa/LSC) denote the junction sites between each corresponding two regions of the genome.

**Figure 6 plants-10-00283-f006:**
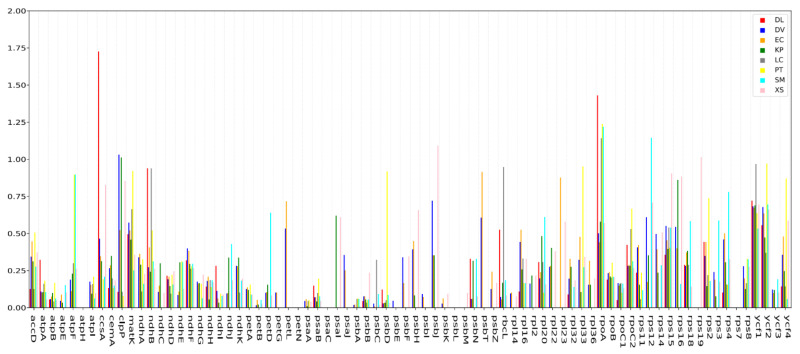
The Ka/Ks ratios of 78 protein-coding genes of the *N. lappaceum* chloroplast genome versus eight closely related species of Sapindaceae.

**Figure 7 plants-10-00283-f007:**
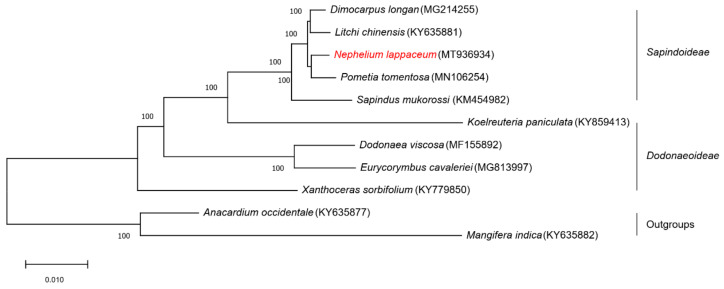
The maximum likelihood (ML) phylogenetic tree of the Sapindaceae family based on chloroplast genome sequences. The numbers in each node were tested by bootstrap analysis with 1000 replicates. *Anacardium occidentale* and *Mangifera indica* were set as the outgroups. The position of *N. lappaceum* is indicated in red text.

**Table 1 plants-10-00283-t001:** Comparison of the general features of the nine Sapindaceae chloroplast genomes.

Genome Feature	*Dimocarpus longan*	*Litchi chinensis*	*Pometia tomentosa*	*Sapindus mukorossi*	*Nephelium lappaceum*	*Dodonaea viscosa*	*Eurycorymbus cavaleriei*	*Koelreuteria paniculata*	*Xanthoceras sorbifolium*
GenBank	MG214255	KY635881	MN106254	KM454982	MT936934	KM454982	MF155892	MG813997	KY859413
Size (bp)	160,833	162,524	160,818	160,481	161,321	159,375	158,777	163,258	161,231
LSC (bp)	85,707	85,750	85,666	85,650	86,068	872,014	86,940	90,236	85,299
SSC (bp)	18,270	16,568	18,360	18,873	18,153	17,972	17,991	18,268	18,692
IR (bp)	28,428	30,103	28,396	27,979	28,550	27,099	26,923	27,377	28,620
Total genes	132	132	133	135	132	135 (2 Pseudogene)	137	133 (3 Pseudogene)	132
Protein genes	87	87	88	88	87	88	89	85	86
tRNA genes	37	37	37	39	37	37	40	37	38
rRNA genes	8	8	8	8	8	8	8	8	8
GC (%)	37.79%	37.80%	37.87%	37.66%	37.77%	37.86%	37.92%	37.30%	37.69%

Note: IR: Inverted repeats. GC: GC content.

## Data Availability

All data generated or analyzed during this study are included in this published article.

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
