# Peer review of "Chloroplast Genome of Rambutan and Comparative Analyses in Sapindaceae"

_plants, 2021, doi:10.3390/plants10020283_

Round 1
Reviewer 1 Report
Manuscript ID: plants-1085712
Title: Chloroplast genome of rambutan and comparative analyses in Sapindaceae
By Fei Dong et al.
Overview: This study reported complete chloroplast (cp) genomes of Rambutan (Nephelium lappaceum L.). The authors presented the cp genomes of this species and performed comparative and phylogenetic analyses with other published related species. English writing is good, and results were well supported by tables, figures, and supplemental materials. In general, this manuscript is quite exciting, and I think this manuscript is an excellent addition to the plant genomics resource. I would like to see the paper published in the ‘plants’. I think minor revisions should be needed for this paper to publish.
Point 1: Please change the family, subfamily names to non-italics in the whole text.
e.g., Sapindaceae > Sapindaceae; Sapindoideae > Sapindoideae
Point 2: Please write down the full genus name and the authority (standard abbreviation) of the scientific name when the authors first mentioned the scientific name in the text.
e.g., P2 L62: Nicotiana tabacum > Nicotiana tabacum L.
P2 L71: Nephelium Linn > Nephelium L.
P3 L87: S. mukorossi > Sapindus mukorossi Gaertn.
P3 L88: P. tomentosa > Pometia tomentosa (Blume) Teijsm. & Binn.
P3 L88: D. Longan > Dimocarpus longan Lour.
P3 L89: L. chinensis > Litchi chinensis Sonn.
Point 3: P3 Figure 1: Please change the authority to standard abbreviation and non-italics.
Nephelium lappaceum Linn > Nephelium lappaceum L.
Point 4: P3 Table 1: Please check the total gene number.
Moreover, why did not mention the differences of tRNA genes of Sapindus mukorossi in studied species? Only Sapindus mukorossi has 39 tRNA genes.
Point 5: P4 L129: Why the authors used 53 proteins? Please check this part.
Point 6: P6 L129: Why the authors compare only four species? I could find chloroplast, the complete genome of another species in Sapindoideae from NCBI (Koelreuteria paniculata (NC_037176.1)). Can these four species represent the subfamily?
Point 7: P8 Figure 6: The diagram in this figure is too small to look at.
Point 8: P8 Figure 7: Why the authors include only four species among previous data in Sapindoideae?
Point 9: P9 L250~261, L262~270: This part is too general. Please discuss the inclusion of this studied result.
Point 10: P9 L285: Please reconsider this word ‘synteny’. How about ‘similarity’?

Reviewer 2 Report
Figure2, what does the green line in Fig 2A mean?
Line 138, Please provide detailed information about the RNA editing sites identification.
Line 144, please provide the data of the frequency of serine codon editing.
Figure 4, please provide the name of the five species of Sapindoideae used for the analysis.
Line 135, There are some grammar mistakes, such as "30 codons with RSCU values more than 1" this sentence is not complete. I strongly encourage the author to have the manuscript to be edited by professional editor before publish.
Round 2
Reviewer 2 Report
The author addressed all my concerns, I recommend it for publishing in Plants.